# Principal investigators over-optimistically forecast scientific and operational outcomes for clinical trials

**Daniel M. Benjamin[1], Spencer P. Hey[2], Amanda MacPherson[3], Yasmina Hachem[3], Kara S. Smith[3], Sean X. Zhang[3], Sandy Wong[3], Samantha Dolter[3], David R. Mandel[4], Jonathan Kimmelman[3]\***

**1** Huizenga College of Business and Entrepreneurship, Nova Southeastern University, Ft Lauderdale, FL, United States of America, **2** Center for Bioethics, Harvard Medical School, Boston, MA, United States of America, **3** Studies in Translation, Ethics and Medicine, Biomedical Ethics Unit, McGill University, Montreal, QC, Canada, **4** York University, Toronto, ON, Canada

\* Jonathan.kimmelman@mcgill.ca

## Abstract

### Objective

To assess the accuracy of principal investigators' (PIs) predictions about three events for their own clinical trials: positivity on trial primary outcomes, successful recruitment and timely trial completion.

### Study design and setting

A short, electronic survey was used to elicit subjective probabilities within seven months of trial registration. When trial results became available, prediction skill was calculated using Brier scores (BS) and compared against uninformative prediction (i.e. predicting 50% all of the time).

### Results

740 PIs returned surveys (16.7% response rate). Predictions on all three events tended to exceed observed event frequency. Averaged PI skill did not surpass uninformative predictions (e.g., BS = 0.25) for primary outcomes (BS = 0.25, 95% CI 0.20, 0.30) and were significantly worse for recruitment and timeline predictions (BS 0.38, 95% CI 0.33, 0.42; BS = 0.52, 95% CI 0.50, 0.55, respectively). PIs showed poor calibration for primary outcome, recruitment, and timelines (calibration index = 0.064, 0.150 and 0.406, respectively), modest discrimination in primary outcome predictions (AUC = 0.76, 95% CI 0.65, 0.85) but minimal discrimination in the other two outcomes (AUC = 0.64, 95% CI 0.57, 0.70; and 0.55, 95% CI 0.47, 0.62, respectively).

### Conclusion

PIs showed overconfidence in favorable outcomes and exhibited limited skill in predicting scientific or operational outcomes for their own trials. They nevertheless showed modest

**Data Availability Statement:** All data used in these analyses have been deposited in Open Science Framework: https://osf.io/67qn4.

**Funding:** Financial support for this study was provided entirely by a grant from Canadian Institute of Health Research (EOG 201303). The funding agreement ensured the authors' independence in designing the study, interpreting the data, writing, and publishing the report. https://cihr-irsc.gc.ca/e/193.html.

**Competing interests:** The authors have declared that no competing interests exist.

ability to discriminate between positive and non-positive trial outcomes. Low survey response rates may limit generalizability.

# 1. Introduction

Clinical trials aim at generating evidence for clinical decision-making. Many trials fail to generate clinically relevant information because they posit a poorly justified hypothesis, they deploy suboptimal design and reporting, or they founder on operational issues like recruitment [1]. Although such failures have many causes, one likely factor is that investigators misjudge the viability of their clinical hypotheses or operational aspects of the trial itself.

Some commentators allege that investigators tend to overestimate efficacy [2], overestimate disease progression in comparator arms [3], underestimate feasibility challenges when designing clinical trials [4] or overestimate benefit during informed consent [5, 6]. However, claims about unrealistic optimism have little direct support [7]. For example, the recurrent use of unrealistically large effect sizes in power calculations is sometimes invoked to suggest investigators harbor excess optimism about clinical hypotheses [8, 9]. However, the resulting underpowered design might instead reflect investigators' realism about their inability to recruit enough patients for larger studies aimed at detecting smaller effects.

While there is a large literature on the quality of expert and physician judgment [10] and propensities toward unrealistic expectations [11], we are aware of only two studies that directly tested investigator judgments in clinical trials [12, 13]. Neither study specifically focused on principal investigators, and neither assessed investigator judgments about operational trial outcomes like recruitment or timely study completion. However, skill with anticipating recruitment or feasibility challenges is crucial for successful implementation and efficient execution of trials.

In what follows, we describe the extent to which PIs harbor unbiased and accurate judgments about the prospects of primary outcome attainment, successful recruitment, and on-time completion of primary data collection for their own clinical trials.

# 2. Methods

## 2.1 Overview

Our basic approach to assessing the quality of judgments of PIs was to ask a series of them to offer subjective probabilities (hereafter called "forecasts") about the attainment of three events in their own trial: statistical significance in favor of the experimental intervention on the primary endpoint, attainment of the projected sample size by trial completion, and actual trial closure by the primary completion date initially projected in clinical trial registration records. When trials were completed, we then analyzed the relationship between PI forecasts and actual outcomes.

## 2.2 Trial sample

We searched ClinicalTrials.gov on a monthly basis between September 24, 2013 and April 24, 2017 to accumulate a sample of newly registered clinical trials. Clinical trials were included if they met the following criteria: 1) registered as not yet recruiting; 2) registration record provided a PI; 3) interventional trial; 4) at least one study site or sponsor registered in one of the six jurisdictions widely regarded as maintaining effective trial oversight apparatuses (i.e. USA,

Canada, a European Medicine's Agency- affiliated or partnered country, Japan, Australia, and New Zealand); this inclusion criterion was intended as a quality control for trials); and 5) minimum sample size (at least 20 patients). In addition, we excluded non-inferiority studies to simplify elicitation. Trial selection was automated, with trials reviewed by research assistants prior to inviting PIs. Inclusion criteria was reassessed during trial character coding.

Basic characteristics of each trial were collected from the registration record on ClinicalTrials.gov. Treatment category was based on intervention classifications from ClinicalTrials. gov. Trials with multiple treatment types were counted in each category. If multiple interventions were tested, the FDA status was counted as approved only if all drug/biologic interventions were approved at the time of testing. If no drugs or biologics were tested in a trial, the FDA status was considered not applicable. In some cases, interventions such as supplements were classified as drugs on ClinicalTrials.gov but considered not applicable for the purpose of FDA approval status.

### 2.3 Forecast elicitation

Within 6.5 months of having first registered a new trial, one PI per trial was sent a survey embedded in an email soliciting forecasts. PIs who actively declined, or who did not participate after two reminder emails, were recorded as having declined participation. PIs were not given any financial incentive for completing our survey.

PIs involved in more than one trial were invited to provide forecasts for only one (their earliest registered trial.).

Surveys elicited four forecasts concerning the *probability* that trials would achieve: 1) statistical significance on their primary outcome, 2) complete recruitment of their registered target enrollment by trial closure, 3) trial closure by the primary completion date projected in first registration record, and 4) exceed a threshold (10% for mono; 15% for combo-therapy) of grade 3 or greater adverse events (we do not report these results due to inconsistencies in how they are reported). Probabilities were elicited by asking for a value between and inclusive of 0% and 100%. Forecasts regarding primary outcome were omitted for healthy volunteer phase 1 trials. Additional information on surveys, and protocol deviations, are described in the **S1 File**.

For phase 1 trials (24 of 740 trials for which primary endpoint forecasts were elicited), the first question was adjusted as follows: we asked about a primary efficacy outcome (n = 8) or where this was not an option, we asked whether the study would identify a maximally tolerated dose five times greater than the study starting dose (n = 16).

Surveys were designed to take no more than five minutes to minimize survey burden and maximize response rates. The entire survey was embedded in the email invitation to increase transparency and response rates. Offline, we recorded the location and h-index of respondents and a random sample of 100 non-respondents, our target number for primary outcome forecasts, to probe response bias matching the primary endpoint sample size we targeted. We also collected trial characteristics (e.g. use of randomization, sample size, etc.) of trials associated with investigators who participated as well as investigators who declined participation. Methods for extracting this demographic information are provided in the **S1 File**.

### 2.4 Verification of events

We determined whether forecasted events had occurred (coded as '1') or had not occurred ('0') as follows: for primary outcome attainment, we sought out publications, results logged on ClinicalTrials.gov, or press releases for each registered trial once it attained a completion status on its primary endpoint. Trials that were closed or terminated early due to efficacy were coded

as '1'; trials closed or terminated early due to futility for reasons other than recruitment were coded as '0'. We obtained actual recruitment and closing date by accessing ClinicalTrials.gov registration records. Trials recruiting ≥100% of their sample, and closing on or before the registered close date, were scored as having attained their objectives. In cases where we were unsuccessful ascertaining outcomes, we emailed investigators directly. Because our three events occur on different timelines (e.g. a trial might be reported long after a trial completes recruitment and is declared as closed), the sample size of forecasts for each event-type vary.

## 2.5 Analysis

Once primary outcomes were available for 100 clinical trials, we stopped data collection and performed analysis (June 12, 2019; four additional outcomes that had been solicited before this date and received by email afterwards were included in our analysis). Since many trials posted final recruitment numbers without posting results, we were able to verify recruitment attainment for more trials than we were for primary endpoint attainment. Similarly, since study status always indicated whether a trial was open or not beyond the initially projected closing date, we had a still larger sample of trials for on-time closure outcomes. The present analysis includes all trials for which outcome data were available. For example, if a trial was registered as open after the date of its projected primary completion date, it was included in our analysis of closing date forecasts, but not in our analyses of recruitment or primary outcome forecasts, since the latter two events had not yet matured. As a sensitivity analysis, we performed all analyses using only the 104 trials for which we had outcomes on all three events (see **Fig A2-A4 in S1 File**) to test if the timing of reporting results is related to their outcomes. Trials that take longer to report results may do so because of problems occurring during trial operations or because positive results are reported more promptly. Our study was not pre-registered as it was launched in 2013, before the pre-registration was the norm for the social sciences.

Our primary metric of prediction skill was the Brier score (BS), a standard measure of prediction accuracy [14]. BS is calculated as the average squared difference between predicted and observed outcomes, where observed trial outcomes are dichotomized based on whether they are positive or not. BS values closer to zero represent greater accuracy. Since each PI only provided one prediction for each outcome and for only one trial, average BS should be understood as the average of a community of investigators, rather than the average of an individual.

To benchmark prediction skill, we compared scores of PIs to those obtained using two simple prediction algorithms that could be used instead of consulting an expert for forecasts. The first, the "uninformative" benchmark, reflects the Brier score that would be obtained if a forecaster always abstained from providing any information regarding whether an event was more likely to occur than not (i.e. always predicting 50%, which would result in BS = 0.25). The second, a "base rate" benchmark, made predictions equal to the base rate for primary outcome positivity, recruitment success, and timely closure for trials [15]. Base rate prediction algorithms draw on historic information, and often provide reasonably accurate, if crude, forecasts [10, 13, 15]. Both algorithms are naïve and easy to implement. We reasoned that if PIs performed significantly worse than either algorithm, trial decision-makers should regard PI forecasts for their own trials with high levels of skepticism. The base rate for primary outcome attainment was approximated by combining the phase graduation probabilities from Hay et al 2014 [16] (this corresponded to 48.3%). The base rate for recruitment was calculated based on 8093 trials registered as not yet recruiting between 2013–2017 that had actual recruitment listed as of December 2019 (54.0%). Base rate for timely closure was calculated based on 7327 trials registered as not yet recruiting between 2013–2017 that had an actual primary completion date listed as of December 2019 (27.3%).

Secondarily, we decomposed BS to assess two other standard and complementary metrics in prediction studies [17, 18]: calibration (how well forecasts matched the relative frequency of observed results) and discrimination (how well forecasts distinguished events that occurred from those that did not). The former was measured using the calibration index [13, 17] as well as whether a z-test for calibration testing whether forecasts are different from observed outcomes [18]; the latter was measured using the area under the curve (AUC) of the receiver operator characteristic plot [19]. Bias and optimism in PI forecasts were assessed by comparing the relationship between the occurrence of actual events and a) mean forecasts and b) optimistic forecasts (i.e. >50%).

We report mean BS with 95% bootstrap confidence intervals; confidence intervals were generated by resampling with replacement over 10,000 iterations. Data analysis was conducted using R statistical software.

## 2.6 Ethics

This study received ethics approval from the McGill University Faculty of Medicine IRB (A09-B43-13A). All investigators provided informed consent within surveys. The funder had no role in the study.

# 3. Results

## 3.1 Sample characteristics

We invited PIs from 4443 unique trials for participation; investigators from 740 trials provided forecasts (16.7% response rate). Of these, 104 had primary outcome data available by June 2019 and were included in the analysis of primary outcome attainment; 281 had closed for recruitment, were terminated, or were withdrawn, and were analyzed for recruitment attainment; and 536 registered earlier closing dates and were included for closing date analysis (**Fig 1**). Characteristics of PI-respondents (and their trials) are provided in **Table 1**, alongside characteristics of non-respondents. Respondents had similar h-indices, and trials used similar endpoints. Trials in our sample for recruitment and close date were very similar to trials for non-respondents. Trials in our sample for primary outcome attainment tended to be larger, later phase, and were more likely to use randomization compared non-respondents.

## 3.2 Forecast characteristics

Histograms representing the forecasts are provided in **Fig 2**. All three histograms indicate a spike at 50%, indicating maximum uncertainty about the forecast event occurring. PIs generally offered forecasts that expressed optimistic expectation (i.e., > 50% probability) that primary outcome, recruitment, and on-time closure would be attained (**Table 2**). Investigator forecasts implied near certainty (i.e. exceeding 90% or under 10%) about primary outcome, recruitment, and completion time in 37%, 62% and 40% of trials, respectively. Forecasts were coarse in granularity. A total of 98%, 97%, and 94%, respectively, of forecasts reported were divisible by five except a few that represent simple fractions (33, 66, 67) or near certainty (1, 98, 99, etc.).

## 3.3 Unrealistic optimism and forecast skill

Mean predictions for primary outcome attainment, recruitment attainment, and on-time closure exceeded the actual occurrence of each of these events (e.g. 59.6%, 78.8%, and 72.2% vs. 43.3%, 42.7%, and 13.4%). Near certainty about achieving each event occurred more frequently than near certainty events would not occur (see Table 2). PIs predicted a greater than 50%

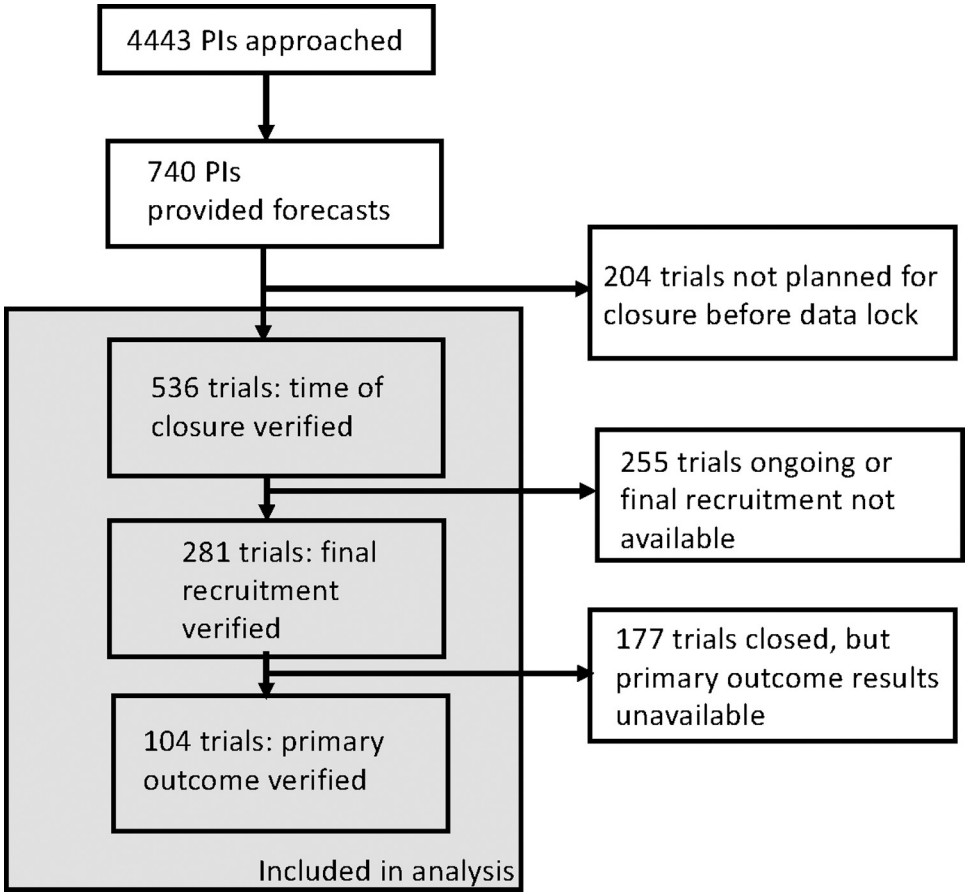

**Fig 1. Participant flow diagram.**

probability of statistical significance on primary outcomes, recruitment attainment, and on-time closure for 58.7%, 83.6% and 79.3%% of clinical trials, respectively. Forecasted events did not materialize in 44%, 52%, and 86% of instances, respectively.

PIs predicted that attainment of primary outcome, recruitment, and on-time closure was more likely than not (i.e. > 50%) for 72%, 90%, and 87% of trials, respectively; these events actually occurred for 56%, 48%, and 14% of trials. The average BS did not surpass the uninformative prediction algorithm or the base rate algorithm for primary outcome (BS = 0.25, 95% CI 0.20, 0.30). PI average BS was significantly worse (higher) than the uninformative algorithm for recruitment attainment and completion date (BS = 0.38, 95% CI 0.33, 0.42; BS = 0.52, 95% CI 0.50, 0.55, respectively) as well as the base rate algorithm (BS = 0.26 and 0.14, respectively; see **Fig A1 in S1 File**).

The PI community showed moderate discrimination skill in primary outcome attainment (AUC = 0.77), but not in recruitment or timeline attainment (**Fig 3**), where perfect discrimination = 1.0 and 0.5 is expected for random guessing. Calibration curves show that PI forecasts exceeded actual outcomes across a wide range of forecast values. PIs showed modest calibration for primary outcome and recruitment attainment, but not for study timelines, where CI is on a squared error-scale with 0 indicating perfect prediction. Nonetheless, experts' forecasts were significantly different than observed outcomes for all three outcomes using Spieglhalter's z-test [18] at p < .001.

**Table 1. Investigator and trial characteristics.** Columns display the various realized outcomes–primary outcome, recruitment, and completion date–and a sample of 100 trials for which PIs did not accept our invitation to participate.

| | | Primary outcome | Recruitment | Completion date | Non-responders |
|---|---|---|---|---|---|
| **Forecaster characteristics** | | N = 104 | N = 281 | N = 536 | N = 100 |
| Location | | | | | |
| | Europe | 40 (38.5%) | 82 (29.2%) | 155 (28.9%) | 23 (23.0%) |
| | N. America | 60 (57.7%) | 184 (65.5%) | 347 (64.7%) | 73 (73.0%) |
| | Asia/Oceania | 1 (1.0%) | 7 (2.5%) | 15 (2.8%) | 4 (4.0%) |
| | Other | 3 (2.9%) | 8 (2.8%) | 19 (3.5%) | 0 (0.0%) |
| H-index | | | | | |
| | median | 22 | 24 | 25 | 25.5 |
| | (95% CI) | (16.3,26.5) | (21.0,26.3) | (23.2,27.3) | (21.3,29.8) |
| | [range] | [1,111] | [1,111] | [0,162] | [1,132] |
| **Trial characteristics** | | | | | |
| Trial Phase[a] | | | | | |
| | Early (phase 1/2) | 64 (61.5%) | 223 (79.4%) | 427 (79.7%) | 80 (80.0%) |
| | Late (phase 3) | 40 (38.5%) | 58 (20.6%) | 109 (20.3%) | 20 (20.0%) |
| Randomization | | | | | |
| | Yes | 83 (79.8%) | 182 (64.8%) | 320 (59.7%) | 60 (60.0%) |
| | No | 21 (20.2%) | 99 (35.2%) | 216 (40.3%) | 40 (40.0%) |
| Indication | | | | | |
| | Cancer | 12 (11.5%) | 64 (22.8%) | 173 (32.3%) | 43 (43.0%) |
| | Non-cancer | 92 (88.5%) | 217 (77.2%) | 363 (67.7%) | 57 (57.0%) |
| Treatment category[b] | | | | | |
| | Drug/biologic | 70 (67.3%) | 196 (69.8%) | 393 (73.3%) | 83 (83.0%) |
| | Supplement | 13 (12.5%) | 34 (12.1%) | 52 (9.7%) | 0 (0.0%) |
| | Device/procedure | 13 (12.5%) | 32 (11.4%) | 77 (14.4%) | 17 (17.0%) |
| | Behavioral | 10 (9.6%) | 27 (9.6%) | 45 (8.4%) | 10 (10.0%) |
| | Other | 6 (5.8%) | 13 (4.6%) | 32 (6.0%) | 5 (5.0%) |
| Sample size | | 67 (10–2000) | 44.5 (5–1500) | 56 (3–2140) | 51 (10–916) |
| | median (range) | | | | |
| Sponsors | | | | | |
| | Commercial | 15 (14.4%) | 33 (11.7%) | 46 (8.6%) | 8 (8.0%) |
| | Non-commercial | 81 (77.9%) | 208 (74.0%) | 399 (74.4%) | 67 (67.0%) |
| | Both | 8 (7.7%) | 40 (14.2%) | 91 (17.0%) | 25 (25.0%) |
| Primary outcome type | | | | | |
| | Efficacy | 78 (75.0%) | 189 (67.3%) | 362 (67.5%) | 66 (66.0%) |
| | Safety | 8 (7.7%) | 50 (17.8%) | 100 (18.7%) | 19 (19.0%) |
| | Preventive | 7 (6.7%) | 16 (5.7%) | 31 (5.8%) | 7 (7.0%) |
| | Other | 11 (10.6%) | 26 (9.3%) | 43 (8.0%) | 8 (8.0%) |
| FDA status at trial launch[c] | | | | | |
| | Approved | 42 (40.4%) | 122 (43.4%) | 248 (46.3%) | 44 (44.0%) |
| | Unapproved | 15 (14.4%) | 51 (18.1%) | 115 (21.5%) | 34 (34.0%) |
| | Not applicable | 47 (45.2%) | 108 (38.4%) | 173 (32.3%) | 22 (22.0%) |
| Date of trial launch (range) | | 06/2013–01/2017 | 06/2013–07/2017 | 05/2013–11/2017 | 07/2013–03/2017 |

[a]Phase 2/3 trials were considered early phase trials.

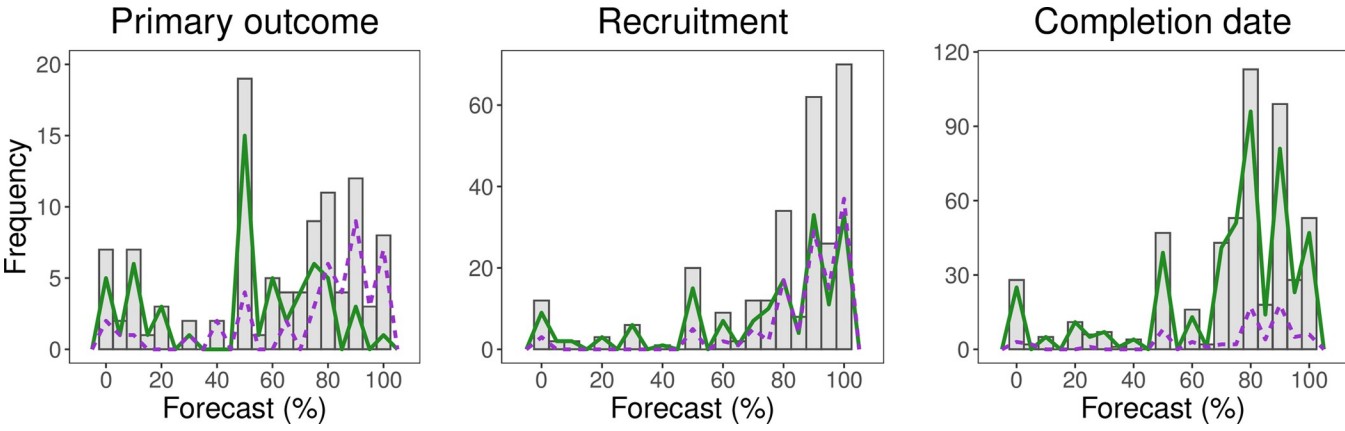

**Fig 2. Histogram of forecasts.** Histogram of forecasts for primary outcome attainment, completion date, and recruitment. Frequency Polygons represent the distribution of forecasts by outcome where the purple, dashed lines are forecasts for positive trials, and the green, solid lines are non–positive trials.

## 4. Discussion

PIs showed consistent patterns of overestimating the prospect of attaining statistical significance on their primary endpoints, recruitment targets and on-time closure. For instance, average forecasts exceeded actual event occurrences, and extremely positive forecasts were far more common than extremely pessimistic forecasts. Calibration curves thus reflect over-optimism for all three outcomes. Of the three events examined in our study, PIs were especially prone to underestimating completion time, reflecting the well-known "planning fallacy" [20].

PIs forecasts were also no more informative than uninformative forecasts (i.e. predicting 50%, which is the equivalent of abstaining from expressing a belief either way). Indeed, in recruitment and timeline attainment, PIs' prediction skills were significantly worse than an uninformative prediction benchmark. PIs nevertheless showed modest ability to discriminate between trials that would produce statistical significance on primary outcome attainment and those that would not.

PI forecasts also raised intriguing questions concerning investigator expectations. First, approximately 6% of PIs expressed extreme doubt their trials would recruit their target sample.

**Table 2. Forecast properties and brier scores.**

| | | Primary outcome | Recruitment | Completion date |
|---|---|---|---|---|
| | | (N = 104) | (N = 281) | (N = 536) |
| **Outcomes (N, %)** | | | | |
| | Positivity | 45 (43.3%) | 120 (42.7%) | 72 (13.4%) |
| **Forecasts** | | | | |
| | Mean | 59.6% | 78.8% | 72.2% |
| | Median | 66% | 90% | 80% |
| | Mode | 50% | 90% | 80% |
| **Extreme forecasts (N, %)** | | | | |
| | < = 10% | 16 (15.4%) | 16 (5.7%) | 35 (6.5%) |
| | > = 90% | 23 (22.1%) | 158 (56.2%) | 180 (33.6%) |
| | **Total** | **39 (37.5%)** | **174 (61.9%)** | **215 (40.1%)** |
| **Brier scores (mean, 95% bootstrap CI)** | | | | |
| | Mean | 0.25 [0.20–0.30] | 0.38 [0.33–0.42] | 0.52 [0.50–0.55] |

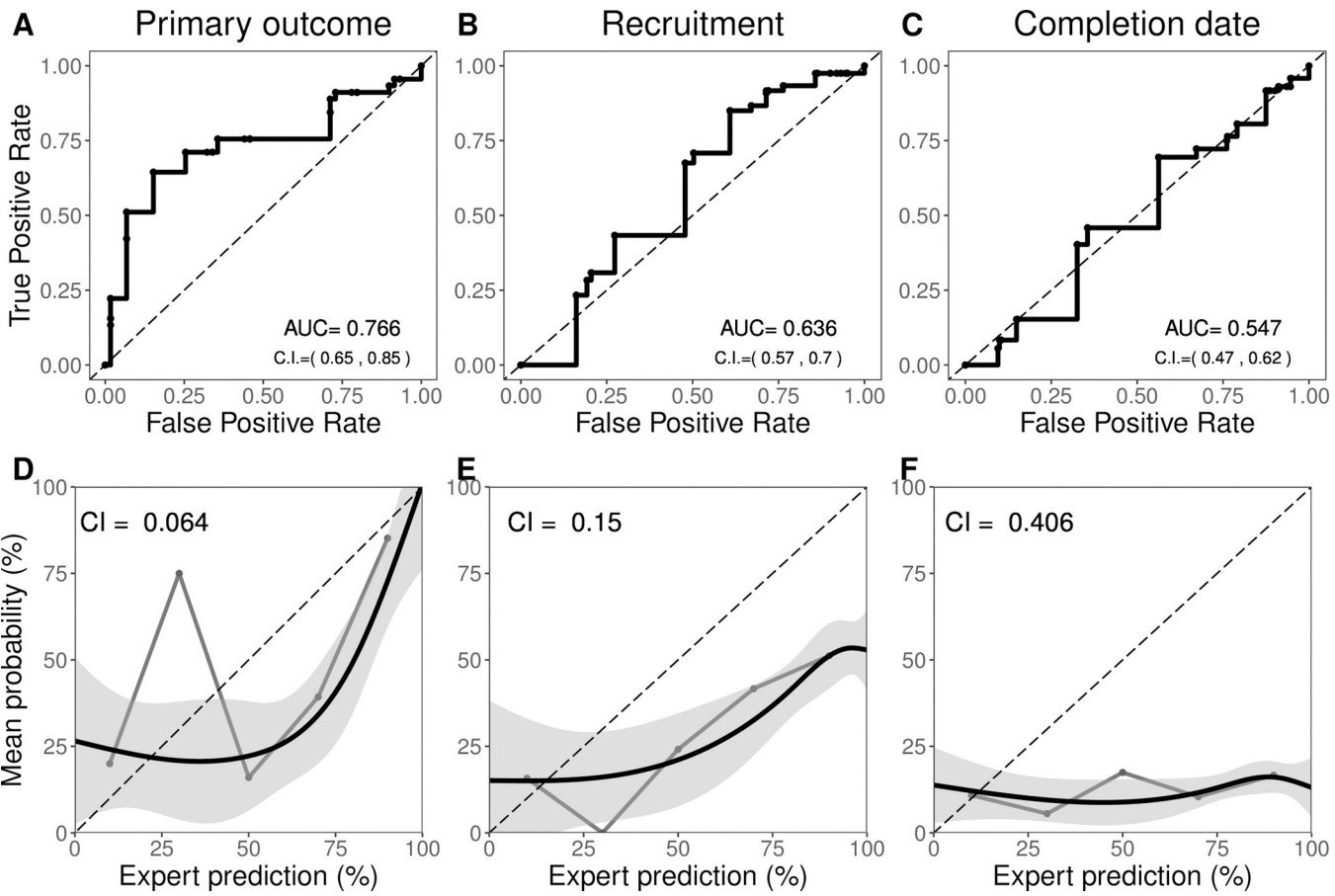

**Fig 3. Discrimination and calibration. A)** Receiver Operator Characteristic. AUC = area under the curve with 95% bootstrap confidence intervals with 2000 resamples. **B)** Calibration Curves. Gray lines show 20%–wide bins, and black lines show model–based (GLM) calibration with confidence region; CI = calibration index.

While such humility is commendable and might motivate a redoubling of effort, it is doubtful investigators should initiate or continue trials they truly believe they are so unlikely to be completed. Under-accrued trials fail to achieve the risk/benefit balance grounding ethical approval [21]. Second, approximately one in five PIs were nearly certain their trial would produce a positive outcome (and one in six were nearly certain it wouldn't). While such expectations are compatible with clinical equipoise, which requires community uncertainty [22], such extremity of expectation may indicate basic misunderstandings about the statistical set-up of clinical trials. For example, if trial protocols posit realistic effect sizes and are designed with 80% power, investigators should rarely offer forecasts exceeding 80%. Our findings suggest that PIs who are extremely pessimistic about operational success show better forecast skill than PIs who offer more moderate forecasts. On the other hand, PIs who are highly confident about operational success show worse forecast skill.

Some of the above findings are consistent with previous studies of prediction in general [10, 11, 23], as well as in clinical care (see, for instance [24–26]). Research prediction has not been well studied. Several studies have suggested that prediction markets involving scientists can be used to forecast whether studies will reproduce [27–29]. In one previous study of trial forecasting, cancer experts showed low skill in predicting primary outcome attainment [13]. In another study, neurologists showed low skill in forecasting the evolution of primary endpoints

in placebo and treatment arms in trials [12]. Both studies also showed tendencies for experts to offer extreme and overconfident predictions. On the other hand, the study of neurologists did not show that co-investigators harbored greater optimism about treatment efficacy than independent experts offering forecasts.

Our findings should be interpreted in light of several limitations. First, low response rates raise concerns about the representativeness of our respondents. For most PI and trial characteristics, however, our respondent sample was similar to a random sample of non-respondents. Each trial characteristic of our non-respondent sample was similar to our full dataset with the greatest deviations (about 10% difference) in disease indication and treatment category. Even if our sample was biased, our findings suggest that important scientific and operational judgments for a meaningful minority of clinical investigators who conduct clinical trials are no more informative than a coin toss. Second, we were unable to verify many efficacy and recruitment forecasts. We cannot rule out that forecasts might have been more accurate for those trials for which we were unable to verify outcomes. Self-selection could also bias response patterns; we speculate that trials reporting results were somewhat biased in terms of primary outcome attainment. If so, our sample may have overestimated the realism and forecast skill of PIs. As regards recruitment and on-time primary completion, a sensitivity analysis that restricted analysis only to those trials for which all three outcomes were available revealed somewhat improved forecast accuracy (see **Fig A2-A4 in S1 File**). This suggests that trials that announce results earlier than expected have smoother operations, which makes them more predictable. Third, the forecasts we elicited may have included judgments beyond outcome likelihood. PIs may have offered optimistic predictions to project confidence. Spikes at 50% for primary outcome attainment may reflect an erroneous perception that PIs should have maximum uncertainty when initiating trials. Another explanation is social desirability bias: PIs may feel it is appropriate to project optimism about their own trials, even if they harbor private doubts. Fourth, poor prediction skill in this study may reflect limited engagement in survey questions. We did not offer PIs incentives to provide accurate predictions. The fact that the vast majority of forecasts used multiples of five despite the option for unit variations in probability may reflect low motivation as highly engaged "superforecasters" tend to make granular distinctions–ie using values such as 68% or 71%—and such granularity in forecasting is predictive of accuracy [30, 31]. Finally, poor forecast skill may reflect difficulty articulating probabilities that reflected actual belief. Future studies should vary elicitation approaches to test measurement invariance. Overall, we urge caution generalizing our findings.

If the patterns observed in this study generalize, they have important implications for research review. Ethics committee or study section members should exercise judicious skepticism for PI testimony regarding treatment efficacy or operational outcome attainment for their own trials. Indeed, they should press for evidence (e.g. supporting or feasibility studies) where PIs offer highly confident claims about their own trials. They should also scrutinize language about direct and indirect benefits in informed consent documents.

Our findings would have similar implications for policy-making. For instance, many recommendations in clinical practice guidelines are based on expert opinion or expert interpretation of low-level evidence [32]. Although expert opinion used in clinical practice guidelines likely entails more considered judgments than those canvassed in our survey, policy makers should recognize the fallibility of expert judgment and seek out independent judgment to corroborate investigators. In the context of cancer care, where treatments have considerable known side effects and costs, clinical practice guidelines might consider restricting the issuance of recommendations based on expert interpretation of low-level evidence.

Finally, our findings potentially have value for principal investigators. Clinical trials aim at detecting signal of treatment efficacy amidst various processes that are unpredictable. These

include stochasticity (e.g. chance imbalances in baseline prognostic variables after randomization) and unexpected events surrounding a trial (who can predict that a totally new treatment strategy emerges during a trial, or that a global pandemic forces investigators to halt recruitment). Clinical trialists would do well to appreciate the extent to which they operate in a world buffeted by unpredictable events. Investigators should also consider contrarian counterfactuals to dampen their optimism. Even experts are not immune to self-deception.

## Supporting information

**S1 File.** 1) Repository for datafiles and codebook, 2) Survey, 3) Fig A1: Histograms of Brier scores, and 4) Sensitivity analysis including Fig A2: Histogram of Forecasts and Brier scores. Fig A3: Calibration curves from sensitivity analysis. Fig A4: discrimination from sensitivity analysis.
(PDF)

## Acknowledgments

We thank Georgie Kourkoupolis and Noga Aharony for additional research assistance. We also thank all the Principal Investigators who participated in our study.

## Author Contributions

**Conceptualization:** David R. Mandel, Jonathan Kimmelman.

**Data curation:** Daniel M. Benjamin, Amanda MacPherson.

**Formal analysis:** Daniel M. Benjamin, Amanda MacPherson.

**Funding acquisition:** Jonathan Kimmelman.

**Investigation:** Yasmina Hachem, Kara S. Smith, Sean X. Zhang, Sandy Wong, Samantha Dolter.

**Methodology:** Daniel M. Benjamin, Spencer P. Hey.

**Project administration:** Amanda MacPherson.

**Software:** Spencer P. Hey.

**Supervision:** Jonathan Kimmelman.

**Visualization:** Daniel M. Benjamin, Amanda MacPherson.

**Writing – original draft:** Jonathan Kimmelman.

**Writing – review & editing:** Daniel M. Benjamin, Spencer P. Hey, Amanda MacPherson, Yasmina Hachem, Kara S. Smith, Sean X. Zhang, Sandy Wong, Samantha Dolter, David R. Mandel.

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
