## [Decision Letter · Decision Letter 0]

16 Oct 2021

PONE-D-21-10921Principal Investigators Over-optimistically Predict Scientific and Operational Outcomes for Their Clinical TrialPLOS ONE

Dear Dr. Kimmelman,

Thank you for submitting your manuscript to PLOS ONE. After careful consideration, we feel that it has merit but does not fully meet PLOS ONE’s publication criteria as it currently stands. Therefore, we invite you to submit a revised version of the manuscript that addresses the points raised during the review process.

First of all, **I would like to thank all 3 reviewers for providing a fast and interesting feedback** on this manuscript. I must acknowledge that it took me some times to secure reviewers for this paper and apologize for the delays. I wanted to have a clear understanding of the statistical approach used and have asked a specific advice to a 3rd reviewer about these aspects. Please try to take into account all the comments made by the 3 reviewers by detailing as much as possible your methods in order to make sure that it will be understandable by a large audience.  

I have **a few additional comments** : 

- In the abstract : please add a few words in the conclusion about the limitations in order to avoid any spin ;

- I appreciate the transparency in the manuscript and in the appendix but I missed a mention of a pre registration : 

. If the study was registered, please provide a specific number of registration ; 

. If the study was not registered, please make it explicit ;

. I would also suggest to edit the method section in order to move any change to the initial protocol in a dedicated paragraph "changes to the initial protocol". 

I agree with the reviewers that the paper is about an important issue. 

We look forward to receiving your revised manuscript.

Kind regards,

Florian Naudet, M.D., M.P.H., Ph.D.

Academic Editor

PLOS ONE

Journal Requirements:

Reviewers' comments:

Reviewer's Responses to Questions

**Comments to the Author**

1. Is the manuscript technically sound, and do the data support the conclusions?

Reviewer #1: Partly

Reviewer #2: Partly

Reviewer #3: Yes

2. Has the statistical analysis been performed appropriately and rigorously? 

Reviewer #1: Yes

Reviewer #2: I Don't Know

Reviewer #3: Yes

3. Have the authors made all data underlying the findings in their manuscript fully available?

Reviewer #1: No

Reviewer #2: No

Reviewer #3: Yes

4. Is the manuscript presented in an intelligible fashion and written in standard English?

Reviewer #1: Yes

Reviewer #2: Yes

Reviewer #3: Yes

5. Review Comments to the Author

Reviewer #1: ***************************************************************************************************

please see the attached report

Reviewer #2: Benjamini and colleagues aimed to assess the accuracy of principal investigators’predictions about three events for their own clinical trials: positivity on trial primary outcomes, successful

recruitment and timely trial completion. The topic is of interest but the manuscript needs major revisions before being considered for publication.

The main remark is the difficulty to read the results due to multiplicity and the unusual statistical methods used. The Brier score is well explained but the utility of the comparison of PI scores to predictions algorithms is difficult to understand. I wonder if a simpler analysis could show similar results. For example, for the primary endpoint, authors could show the percentage of studies for which PIs expected statistical significance and finally were negative.

Table 1: why adding columns 2 and 3 (with respective n= 281 and 536)? What is their utility regarding the aim of the study? In the same way why showing column 4 (match sample of 100 trials). This is confusing for me; the footnote does not mention the origin of the different columns.

Fig 3 needs clarifications. At least, foot notes should be added. The results section says that when referring to figure 3, PI showed moderate discrimination for primary outcome but not for recruitment or timeline attainment. Please explain how you can say that when looking at the figure.

The utility of sensitivity analysis needs also to be clarified, especially the ROC curves shown in fig A3.

Reviewer #3: The article is interesting and well-written. However, a think there is the room for a further improvement. In particular, it is well-known that there are several decompositions of the Brier score which provide a deeper insight. Therefore, I wonder if one of these decompositions may be useful to obtain further insights on the data analyzed by the authors. Moreover, I have the additional minor remarks:

1. Captions of tables and figures are too long.

2. In sentences like “timely closure for trials.(15)”, “metrics in prediction studies:(16) calibration”, and “measured using the calibration index;(13)(16) the”, the numbered citations should be placed before the punctuation.

6. PLOS authors have the option to publish the peer review history of their article (what does this mean?). If published, this will include your full peer review and any attached files.

Reviewer #1: **Yes: **Leonhard Held

Reviewer #2: No

Reviewer #3: No

---

## [Decision Letter · Decision Letter 1]

23 Dec 2021

PONE-D-21-10921R1Principal Investigators Over-optimistically Forecast Scientific and Operational Outcomes for Clinical TrialsPLOS ONE

Dear Dr. Kimmelman,

Thank you for submitting your manuscript to PLOS ONE. After careful consideration, we feel that it has merit but does not fully meet PLOS ONE’s publication criteria as it currently stands. Therefore, we invite you to submit a revised version of the manuscript that addresses the points raised during the review process.

I still think that this is an important paper, thank you for submitting it. I really appreciate your edits.

**I would like to thank the 3 reviewers**. As you will see they were pleased with your edits. One of the reviewers still raise a few comments. I must say that I agree with him and I'm looking forward to reading your responses. 

We look forward to receiving your revised manuscript.

Kind regards,

Florian Naudet, M.D., M.P.H., Ph.D.

Academic Editor

PLOS ONE

Journal Requirements:

Reviewers' comments:

Reviewer's Responses to Questions

**Comments to the Author**

1. If the authors have adequately addressed your comments raised in a previous round of review and you feel that this manuscript is now acceptable for publication, you may indicate that here to bypass the “Comments to the Author” section, enter your conflict of interest statement in the “Confidential to Editor” section, and submit your "Accept" recommendation.

Reviewer #1: (No Response)

Reviewer #2: All comments have been addressed

Reviewer #3: All comments have been addressed

2. Is the manuscript technically sound, and do the data support the conclusions?

Reviewer #1: Yes

Reviewer #2: Yes

Reviewer #3: Yes

3. Has the statistical analysis been performed appropriately and rigorously? 

Reviewer #1: Yes

Reviewer #2: Yes

Reviewer #3: Yes

4. Have the authors made all data underlying the findings in their manuscript fully available?

Reviewer #1: (No Response)

Reviewer #2: Yes

Reviewer #3: Yes

5. Is the manuscript presented in an intelligible fashion and written in standard English?

Reviewer #1: Yes

Reviewer #2: Yes

Reviewer #3: Yes

6. Review Comments to the Author

Reviewer #1: Comments on the response to my comments on PONE-D-21-10921. The

numbers refer to the order of my original comments.

1) Thanks for giving more details on the elicitation procedure. Your

explanation indicates the presence of digit preference bias with all

percentages divisible by 5, except for 1, 99, 33 and 67. This should

be mentioned as a limitation of the study.

3) Thank you for your explanation, I was inaccurate in my point. Yes,

the probability of a significant (positive) finding is

Pr(sig) = Pr(sig | H0) Pr (H0) + Pr(sig | H1) Pr(H1)

Now Pr(sig | H0) = usually 5% and Pr(sig | H1) is the assumed power

(say 80%). Therefore, if the clinician reports Pr(sig) it is possible

to infer her personal prior probability Pr(H0) (or equivalently Pr(H1)

= 1 - Pr(H0)). For example, if Pr(sig)=70% is reported by the

clinician we obtain Pr(H0) = 13% and so Pr(H1)=87%. Not sure if this

is helpful for the manuscript but I just wanted to make my point clear.

Minor point "Brier scores": Maybe I wasn't clear but I still think it

does not make sense to report Brier scores of extreme forecasts

because, under the null hypothesis of perfect calibration, the

expectation of the mean BS is (sum p_i*(1-p_i))/n, see Spiegelhalter

(1984, p. 427). If you only look at extreme forecasts (p_i very small

or very large), the mean BS will (under H0) therefore be on average

smaller than overall.

It seems that you need to enlarge the upper limit of ylim in Fig 2, as

one bar currently touches (or crosses?) the limit.

General comment: The absence of a study protocol is a weak point. How

can we criticize the design and reporting standards of clinical trials

if our own standards are not any better? Surely, study protocols have

been very common in medical research prior to 2013. Not just for

clinical trials, but also for other study types (e.g. PRISMA was first

published in 2009).

Reviewer #2: I would like to thank the authors for their responses to reviewers comments and modifications that clarifed the manuscript and allow now its publication.

Reviewer #3: The authors did a good job in revising the paper according to the received comments. Therefore, the paper can be accepted as it is.

7. PLOS authors have the option to publish the peer review history of their article (what does this mean?). If published, this will include your full peer review and any attached files.

Reviewer #1: No

Reviewer #2: **Yes: **Bruno Laviolle

Reviewer #3: No

---

## [Author Response · Author response to Decision Letter 1]

4 Jan 2022

I am puzzled why PLoS One has this box when it also requires we upload a document containing a referee response- a suggestion to the journal for fixing this bug in their submission system ; )

Anyway I pasted this below:

PONE-D-21-10921R1

Dear Editor and Reviewer:

We appreciate the editor and referee offering suggestions to hone the analyses and messaging. See revisions below.

Reviewer #1: Comments on the response to my comments on PONE-D-21-10921. The

numbers refer to the order of my original comments.

1) Thanks for giving more details on the elicitation procedure. Your

explanation indicates the presence of digit preference bias with all

percentages divisible by 5, except for 1, 99, 33 and 67. This should

be mentioned as a limitation of the study.

>>We added a description of forecast granularity as a result in section 3.2 (Forecast Characteristics). We added further discussion of this finding in the limitations section of the Discussion. In this setting, we find this level of granularity is consistent with a lack of motivation of forecasters. We add references to Mellers et al (2015) and Tetlock & Gardner (2016) supporting the association between granularity and accuracy.

3) Thank you for your explanation, I was inaccurate in my point. Yes,

the probability of a significant (positive) finding is

Pr(sig) = Pr(sig | H0) Pr (H0) + Pr(sig | H1) Pr(H1)

Now Pr(sig | H0) = usually 5% and Pr(sig | H1) is the assumed power

(say 80%). Therefore, if the clinician reports Pr(sig) it is possible

to infer her personal prior probability Pr(H0) (or equivalently Pr(H1)

= 1 - Pr(H0)). For example, if Pr(sig)=70% is reported by the

clinician we obtain Pr(H0) = 13% and so Pr(H1)=87%. Not sure if this

is helpful for the manuscript but I just wanted to make my point clear.

>>Interesting. Thank you for your clarification. I take it this assumes our experts are coherent forecasters, of course (often not the case!)

Minor point "Brier scores": Maybe I wasn't clear but I still think it

does not make sense to report Brier scores of extreme forecasts

because, under the null hypothesis of perfect calibration, the

expectation of the mean BS is (sum p_i*(1-p_i))/n, see Spiegelhalter

(1984, p. 427). If you only look at extreme forecasts (p_i very small

or very large), the mean BS will (under H0) therefore be on average

smaller than overall.

>> We have removed these rows from Table 2. 

It seems that you need to enlarge the upper limit of ylim in Fig 2, as

one bar currently touches (or crosses?) the limit.

>> We have fixed the margins of Fig2.

General comment: The absence of a study protocol is a weak point. How

can we criticize the design and reporting standards of clinical trials

if our own standards are not any better? Surely, study protocols have

been very common in medical research prior to 2013. Not just for

clinical trials, but also for other study types (e.g. PRISMA was first

published in 2009).

>>To be clear, we had a study protocol. This was/is required for IRB submission. We did not pre-register the hypotheses of this study, which is now fairly common practice in the social sciences, but was not at the time. 

Reviewer #2: I would like to thank the authors for their responses to reviewers comments and modifications that clarifed the manuscript and allow now its publication.

Reviewer #3: The authors did a good job in revising the paper according to the received comments. Therefore, the paper can be accepted as it is.

7. PLOS authors have the option to publish the peer review history of their article (what does this mean?). If published, this will include your full peer review and any attached files.

>> We’re fine with this.

---

## [Decision Letter · Decision Letter 2]

7 Jan 2022

Principal Investigators Over-Optimistically Forecast Scientific and Operational Outcomes for Their Clinical Trials

PONE-D-21-10921R2

Dear Dr. Kimmelman,

We’re pleased to inform you that your manuscript has been judged scientifically suitable for publication and will be formally accepted for publication once it meets all outstanding technical requirements.

**First I would like to thank the reviewer for his super fast feedback.** 

**And importantly, kudos for this nice piece of work that I am pleased to accept for publication.**

Kind regards,

Florian Naudet, M.D., M.P.H., Ph.D.

Academic Editor

PLOS ONE

Additional Editor Comments (optional):

Reviewers' comments:

Reviewer's Responses to Questions

**Comments to the Author**

1. If the authors have adequately addressed your comments raised in a previous round of review and you feel that this manuscript is now acceptable for publication, you may indicate that here to bypass the “Comments to the Author” section, enter your conflict of interest statement in the “Confidential to Editor” section, and submit your "Accept" recommendation.

Reviewer #1: All comments have been addressed

2. Is the manuscript technically sound, and do the data support the conclusions?

Reviewer #1: (No Response)

3. Has the statistical analysis been performed appropriately and rigorously? 

Reviewer #1: (No Response)

4. Have the authors made all data underlying the findings in their manuscript fully available?

Reviewer #1: (No Response)

5. Is the manuscript presented in an intelligible fashion and written in standard English?

Reviewer #1: (No Response)

6. Review Comments to the Author

Reviewer #1: (No Response)

7. PLOS authors have the option to publish the peer review history of their article (what does this mean?). If published, this will include your full peer review and any attached files.

Reviewer #1: **Yes: **Leonhard Held

---

## [Editor Report · Acceptance letter]

30 Jan 2022

PONE-D-21-10921R2 

Principal Investigators Over-Optimistically Forecast Scientific and Operational Outcomes for Clinical Trials 

Dear Dr. Kimmelman:

I'm pleased to inform you that your manuscript has been deemed suitable for publication in PLOS ONE. Congratulations! Your manuscript is now with our production department. 

Kind regards, 

on behalf of

Pr. Florian Naudet 

Academic Editor

PLOS ONE